# SimFLi: Simple Few-Shot Linear Modeling for On-Device LLM Latency Profiling

## Abstract

On-device inference of large language models (LLMs) is increasingly central to mobile and edge AI, yet profiling their latency remains challenging: existing methods are often server-centric, rely on operator-level instrumentation, or incur overheads that make them impractical for constrained devices. We present **Simple Few-Shot Linear modeling (SimFLi)**, a lightweight and training-free profiler that decomposes inference into prefill (time-to-first-token) and decode phases, and estimates latency from only a few token-length probes. Despite its simplicity, SimFLi achieves accurate latency surfaces without requiring training, extensive measurement, or intrusive instrumentation. Across diverse devices and compact LLMs, SimFLi provides consistently strong $R^2$, RMSE, and MAE under a strict few-point budget, reducing measurement cost by more than an order of magnitude compared to baselines. In practice, SimFLi offers a practical, low-overhead tool to guide model, quantization, and backend choices in real-world on-device deployments.

## 1 Introduction

In recent years, large language models (LLMs) such as GPT, Gemini, and Grok have achieved remarkable advances in natural language understanding and generation. These improvements have been largely driven by scaling the number of parameters and optimizing architectures, resulting in substantial leaps in accuracy and generality. However, this progress has also been accompanied by a dramatic growth in model size and computational demands, making the direct deployment of LLMs on resource-constrained devices—such as smartphones, tablets, and laptops—extremely challenging.

The literature consistently highlights this deployment barrier. Prior work has noted that the substantial memory and compute requirements of LLMs restrict their usability on mobile and edge devices, where both power budgets and hardware capacities are tightly constrained (Girija et al., 2025; Xu et al., 2024; Lamaakal et al., 2025). As a result, running full-scale LLMs locally is effectively infeasible, raising an urgent need for techniques that adapt LLM inference to constrained environments.

In response, major research groups and industry players have pursued lightweight alternatives, developing small- and medium-scale language models (SLMs/MLMs). Examples include the Qwen series with various quantized versions, Meta's Llama family (7B–8B distilled and quantized models), and Microsoft's Phi models tailored for low-resource settings (Li et al., 2024b). While these efforts provide a spectrum of model sizes, reducing parameters alone is insufficient. The actual performance of on-device inference—latency, throughput, and efficiency—depends not only on the model but also critically on the device hardware (Xiao et al., 2024; Li et al., 2024a). Consequently, accurate measurement tools are indispensable for assessing real-world deployability.

Profiling is particularly important in mobile and edge environments, where Size, Weight, and Power (SWaP) constraints dictate the practicality of deployment (Tian et al., 2025). Running LLMs locally can deliver unique benefits such as stronger data privacy and reduced response latency, but also poses the fundamental challenge of balancing model quality with hardware-constrained latency and throughput (Li et al., 2024c). To address this challenge, one requires profiling methods that are both precise and lightweight.

Yet, existing profiling approaches remain limited. Most studies have focused on server-class environments, while on-device profiling has largely targeted CNNs/DNNs (Chu et al., 2023; Zhang et al., 2021) or relied on operator-level measurements that fail to capture LLM-specific dynamics such as time-to-first-token (TTFT), tokens-per-output-time (TPOT), and the role of KV-cache. Alternative approaches—such as neural architecture search (NAS)-based predictors (Li et al., 2023; Akhauri & Abdelfattah, 2024; Feng et al., 2024; Lee et al., 2021) or exhaustive operator-level benchmarking (Li et al., 2024d)—are typically too costly in terms of data collection and training, making them impractical for the immediacy of mobile use cases.

A key insight from recent system research is that LLM latency decomposes naturally into two distinct phases: *prefill* and *decode*. As highlighted by DistServe (Zhong et al., 2024) and vLLM (Kwon et al., 2023), the prefill phase involves prompt tokenization, KV-cache construction, and the generation of the first output token, whereas the decode phase autoregressively generates subsequent tokens. This decomposition suggests a pathway for profiling that avoids exhaustive operator-level instrumentation.

Building on this observation, we propose a **train-free, lightweight profiling methodology** based on *few-shot plotting*. Our approach requires measuring only a small number of carefully chosen input/output token-length combinations, from which we infer the entire latency curve. This method bypasses the need for operator-level dependency, data-intensive training, or costly profiling procedures.

In summary, our contributions are as follows:

- We establish the necessity of **device-tailored latency profiling** for LLM deployment in resource-constrained settings.
- We introduce **Simple Few-Shot Lining (SimFLi)**, a novel profiling method that is train-free, lightweight, and operator-agnostic.
- Our SimFLi demonstrates that only input and output data we can adjust profiler to every combination of device, framework and model
- We demonstrate that SimFLi achieves **high accuracy at low cost**, outperforming baseline approaches in profiling efficiency and precision and even lower bound of latency shows higher regression scores.

## 2 BACKGROUND

### 2.1 EXISTING LATENCY PROFILING APPROACHES

**On-device latency profiling.** Most existing profiling efforts target either device-level deployment of CNN/DNN models or server-side deployment of LLMs (Dhar et al., 2021; Wess, 2025; Chu et al., 2023; Agrawal et al., 2024) . For instance, *nnPerf* (Chu et al., 2023) profiles mobile platforms by collecting operator- and kernel-level traces, but its design remains specialized for feedforward vision models rather than autoregressive LLMs. Consequently, it fails to capture critical LLM-specific dynamics such as prefill/decode asymmetry, time-to-first-token (TTFT), tokens-per-output-time (TPOT), and KV-cache growth. Furthermore, operator-level profiling (Li et al., 2024d) typically depends on hooks from execution frameworks (e.g., TensorFlow Lite), which introduce significant runtime overhead and limit applicability for continuous, online decision-making. (Chu et al., 2023; Yousefzadeh-Asl-Miandoab et al., 2023)

**Kernel-Level predictors.** Another line of work decomposes inference into kernel- or operator-level units and learns predictors from collected latency data. (Chu et al., 2023; Agrawal et al., 2024) For example, *nn-METER* (Zhang et al., 2021) fits device-specific predictors at the kernel granularity, achieving improved accuracy. However, such predictors are highly sensitive to hardware, framework, and kernel fusion choices, all of which vary across platforms (Li et al., 2023). This dependence hinders generalization. In addition, data collection requires device-specific calibration, which is costly to maintain. Critically, autoregressive decoding dynamics unique to LLMs—such as TPOT behavior and cache accumulation—are not directly modeled. Reliance on external profilers further exacerbates runtime and memory overheads, making them unsuitable for mobile deployment (Li et al., 2024d).

**NAS-based predictors.** Latency predictors developed for neural architecture search (NAS) employ data-driven regression or meta-learning approaches to achieve cross-device transferability. LitePred (Feng et al., 2024) builds operator-level datasets and trains MLP prediction with transfer learning for efficient adaptation to new devices, while HELP (Lee et al., 2021) applies meta-learning to improve prediction under unseen hardware. These methods, however, still rely on a few-shot calibration data per device. More importantly, their predictors are designed for CNN/MLP operators, leaving LLM-specific operations (attention, KV-cache access, prefill/decode imbalance) underrepresented. Meta-learning approaches also incur substantial training cost, reducing suitability for fast and lightweight profiling.

**Roofline-based approaches.** Recently, roofline models have been adapted for LLM latency estimation (Imai et al., 2024), combining roofline-based resource bounds with regression models. While these methods provide insight into performance ceilings, they largely focus on server-class GPUs and leverage performance counters available in NVIDIA platforms. Such assumptions limit their applicability to mobile SoCs, where hardware observability and runtime characteristics differ significantly.

## 2.2 PROBLEM STATEMENT

Across these categories, three major gaps emerge. (i) There is a lack of lightweight, LLM-specific on-device profilers that capture autoregressive latency characteristics such as TTFT and TPOT. (ii) Kernel- and operator-level approaches not only heavy (Zhang et al., 2021) suffer from dependence on vendor-specific optimizations (Zhang et al., 2021), complicating generalization across devices (Li et al., 2023), it is really diffcult to build "universally" well-adpated and lightweight model. (iii) Data-intensive approaches, including NAS-based predictors and roofline regressions Imai et al. (2024), require device-specific calibration or rely on coarse simplifications, leaving a gap between accuracy and deployability. Roofline regression, which is our baseline, shows that from in-domain to out-domain test, they need to know model and server information and combine those data to get satisfiable result.

These limitations point to the need for a profiler that is: (i) Well reflecting LLM's features: Prefill and Decode phase (ii) train-free or minimally reliant on device-specific calibration, directly aligned with prefill/decode latency decomposition, and (iii) not knowing device information, yet with a few points, figuring it out indirectly. Our work addresses this need.

## 3 PRELIMINARY EXPERIMENTS

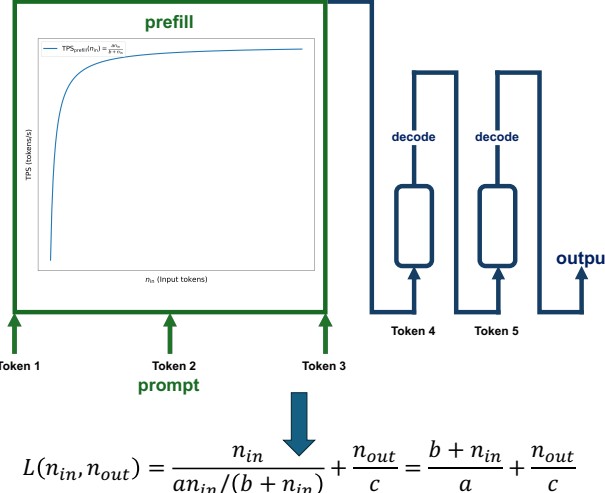

$$L(n_{in}, n_{out}) = \frac{n_{in}}{an_{in}/(b + n_{in})} + \frac{n_{out}}{c} = \frac{b + n_{in}}{a} + \frac{n_{out}}{c}$$

Figure 1: Overview of the proposed method.

## 3.1 MOTIVATION

Prior work on large language model (LLM) inference latency shows that total latency can be decomposed into two phases: *prefill* and *decode* (Imai et al., 2024). The prefill stage grows with the number of input tokens (Li et al., 2024b), whereas the decode stage scales approximately linearly with the number of output tokens (Li et al., 2024b). Existing measurements typically focus on a single framework or device; here we systematically test whether these trends persist across heterogeneous hardware and backends.

In this preliminary study, we measure latency across multiple models, devices, and frameworks to verify the two-phase behavior and to generalize these patterns so that phase-wise latency can be used to build a light yet accurate profiler.

## 3.2 SETUP

**Hardware platforms.** We evaluate on representative smartphones, tablets, and laptops to capture vendor diversity.

Table 1: Hardware platforms used in experiments.

| Category | Model | Environment |
|---|---|---|
| Smartphone | iPhone 14 Pro (A16 Bionic) | iOS app wrapper |
| Smartphone | Galaxy S25 Ultra | Android `adb shell` |
| Tablet | iPad 10 | iOS app wrapper |
| Laptop | MacBook Pro (M1 Pro) | Shell |

For compatibility, Android inference is launched via `adb shell`; iOS latency is measured through a custom Xcode application wrapper (shell access is restricted); laptop runs execute directly in the shell.

**Models and frameworks.** We compare two widely used backends, `llama.cpp` (Gerganov & contributors, 2023) and `MLC-LLM` (Team, 2023). Models are **Qwen3-0.6B** and **Phi-2**, using quantized variants suitable for mobile: `Qwen/Qwen3-0.6B-Q8_0.gguf` and `TheBloke/phi-2-Q4_K_M.gguf` for `llama.cpp` (Team, 2024c; TheBloke, 2024), and `mlc-ai/qwen3-0.6b-q4f16_1-MLC`, `mlc-ai/phi-2-q4f16_1-MLC` for `MLC-LLM` (Team, 2024b;a). We cite both model artifacts and framework repositories for reproducibility.

**Metrics.** We report **TTFT** (prefill latency), **decode latency** (= total − prefill), and **total latency**. TTFT measures time from prompt submission to the first token, including KV-cache construction. Decode latency isolates the incremental generation cost after initialization. Total latency is end-to-end time until the final token.

Further measurement details (repetitions, cooldown, framework overheads) are in Appendix B.

**Experimental protocol.** Latency is collected on a 2D token grid: $\mathcal{S}_{\text{in}} = \{8, 16, \ldots, 120\}$ and $\mathcal{S}_{\text{out}} = \{8, 16, \ldots, 120\}$ (step 8, 15 levels each). We fix context length to 512 and set `MLC-LLM` prefill chunk size to 128 (larger settings were infeasible on targets; Appendix A.1). Each $(n_{\text{in}}, n_{\text{out}})$ pair is evaluated 5 times (randomized order) with a 5-second cooldown. Dataset tokenizers ensure exact token counts.

## 3.3 RESULTS AND FINDINGS

We organize results by backend–device combination (e.g., Qwen3-0.6B with `llama.cpp` on iPhone 14 Pro; Qwen3-0.6B with `MLC-LLM` on iPhone 14 Pro; Qwen3-0.6B with `llama.cpp` on Galaxy S25 Ultra). Figures demonstrate that decode latency grows linearly with output tokens, while TTFT scales with input length. Additional figures and tables appear in Appendix C.

**Results.**   With fixed output (e.g., $n_{\text{out}}{=}16$; Fig. 2d), decode latency is nearly flat over input lengths, while prefill latency increases with input size; total latency follows prefill. With fixed input (e.g., $n_{\text{in}}{=}16$; Fig. 2a), prefill is almost constant, whereas decode latency grows linearly with output length; total latency has approximately the same slope as decode and an intercept equal to TTFT. Prefill shows diminishing increments (saturating throughput) at larger inputs, supporting our two-phase assumptions.

**Implications for few-shot profiling.**   In fixed-input plots, the slope gives the inverse decode TPS and the intercept gives TTFT, so two output lengths suffice to characterize decode. In fixed-output plots, a few input lengths recover a saturating prefill TPS curve (Section 4), enabling accurate surface prediction without exhaustive sweeps. We report means over five repetitions and omit error bars for clarity; variability is small (Table 2).

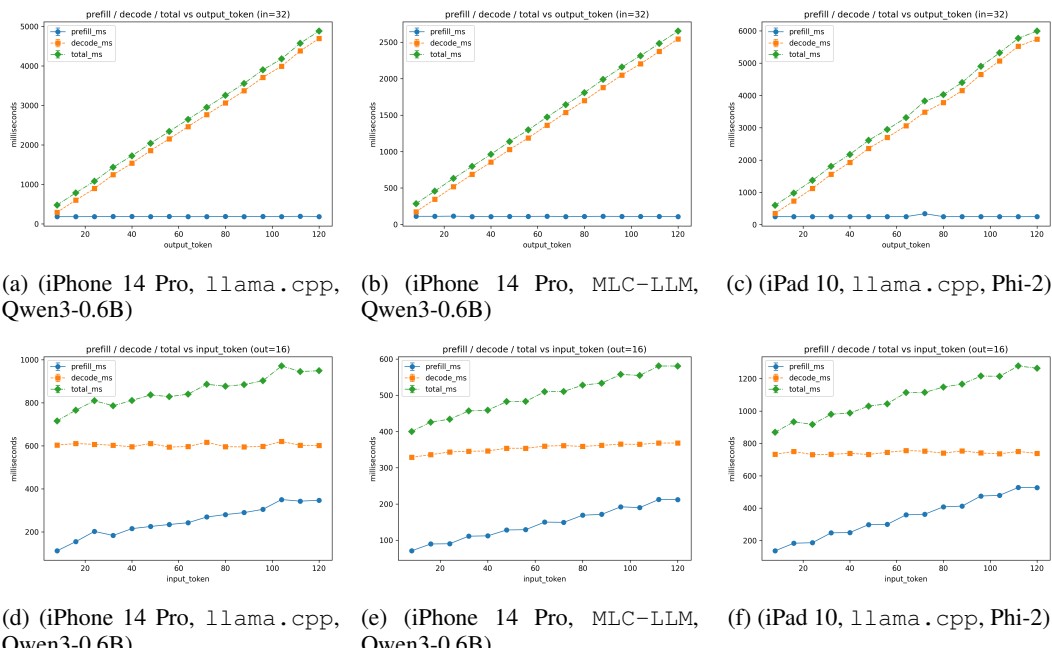

(a) (iPhone 14 Pro, `llama.cpp`, Qwen3-0.6B)

(b) (iPhone 14 Pro, `MLC-LLM`, Qwen3-0.6B)

(c) (iPad 10, `llama.cpp`, Phi-2)

(d) (iPhone 14 Pro, `llama.cpp`, Qwen3-0.6B)

(e) (iPhone 14 Pro, `MLC-LLM`, Qwen3-0.6B)

(f) (iPad 10, `llama.cpp`, Phi-2)

Figure 2: Prefill vs. decode tendencies. (Top: fixed input; bottom: fixed output.)

## 4 METHODOLOGY

### 4.1 LINEAR MODELING

Motivated by Figure 2d, we hypothesize that latency can be expressed as a simple function of input and output token lengths. Since both prefill and decode are naturally measured in tokens, we couple latency with throughput (tokens per second, TPS). Specifically, we select five measurement points, compute the reciprocal of TPS at each phase, and average them to obtain phase coefficients. This yields the following latency model:

$$\text{coef}_{\text{phase}} = \frac{1}{m} \sum_{i=1}^{m} \frac{1}{\text{TPS}_{\text{phase}}(s_i)}, \tag{1}$$

$$L(n_{\text{in}}, n_{\text{out}}) \approx \text{coef}_{\text{prefill}} \cdot n_{\text{in}} + \text{coef}_{\text{decode}} \cdot n_{\text{out}} + \text{C}. \tag{2}$$

where m denotes few-shot estimates. This formulation have already shown strong performance against baselines (Table 3).

## 4.2 SATURATION-AWARE MODELING.

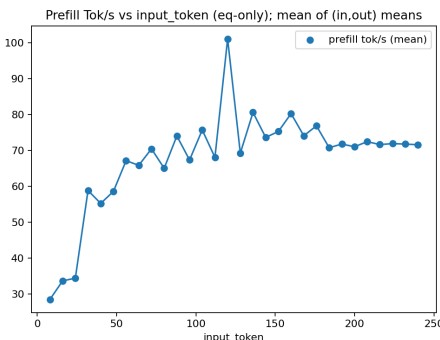

(a) iPhone 14 Pro, llama.cpp, Qwen3-0.6b

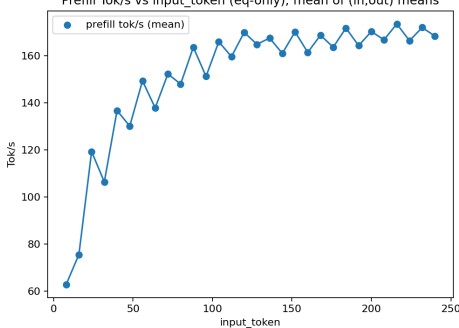

(b) iPhone 14 Pro, MLC-LLM, Phi-2

Figure 3: Prefill TPS shows a trend of becoming more consistent.

Because we decide to utilize the tps, it is inevitable to analyze the tendency of TPS. Figure 3. shows that prefill TPS tends to increase and become flat at some point, which we call it as saturation point. Our previous model, SimFLi1 does not satisfy this condition, which only uses harmonic mean(eq.2) of TPS at some points. For more accurate fitting, it is required to reflect this trend, and we determine the prefill TPS as a function influenced by input token and somewhere converging(eq. 3), which leads to equation(4). However, decode TPS is consistent (see Table.2). The variance and standard deviation(a.k.a std) proves that TPS at decode phase is so steady that we can put it as constant value, which can be evaluated by selecting few point.(eq.2)

$$\text{TPS}_{\text{prefill}}(n) = \frac{a\,n}{b + n}, \qquad \text{TPS}_{\text{decode}} = c, \tag{3}$$

$$L(n_{\text{in}}, n_{\text{out}}) \approx \frac{n_{\text{in}}}{an_{\text{in}}/(b + n_{\text{in}})} + \frac{n_{\text{out}}}{c} = \frac{b + n_{\text{in}}}{a} + \frac{n_{\text{out}}}{c} + \text{C}. \tag{4}$$

Table 2: Decode TPS stability across sampled runs (mean, variance, std).

| Combination | Mean | var | std |
|---|---|---|---|
| iPhone 14 Pro / llama.cpp / Qwen3-0.6B | 26.055 | 0.323 | 0.57 |
| iPhone 14 Pro / MLC-LLM / Phi-2 | 19.092 | 0.432 | 0.657 |
| MacBook M1 Pro / llama.cpp / Phi-2 | 34.988 | 6.854 | 2.618 |

## 4.3 FINDING SATURATION POINTS

Further, we can have a heuristic that if prefill tps goes steady at saturation point, the ratio that decode phase addresses would get larger and larger till saturation point comes. This means that how to choose shots for line fitting is also very important. The point is how to find saturation point using input prompt length. we propose **Few-shot Point selection (FSP)**, which explicitly targets the prefill saturation boundary. The algorithm traces from the maximum context length, halves until a saturation drop is detected, and then symmetrically allocates $k$ points around the boundary (Algorithm 1). By combining this with the saturation model, SimFLi3 consistently outperforms other variants, maintaining $R^2 > 0.80$ even under memory-compressed datasets (Appendix D).

## 5 EVALUATION

### 5.1 BASELINES

**Roofline predictor.** The original roofline predictor (Imai et al., 2024) is designed for server settings; to make it applicable to on-device inference, we incorporate device descriptors from our

---

**Algorithm 1** FSP halving trace, drop test, symmetric selection

---

**Require:** grid $\mathcal{G}$; unique inputs $S$; budget $k \geq 3$; drop threshold $\tau$ (e.g., 0.10)
1: $cur \leftarrow \max S$; $T \leftarrow [\,]$; $boundary \leftarrow \bot$
2: **while** $cur \geq \min S$ **do**                             ▷ Halving trace from endpoint
3:      $s \leftarrow$ nearest element of $S$ to $cur$; $tps(s) \leftarrow s/\text{mean\_prefill\_sec}(s)$
4:      **if** $T \neq \emptyset$ **and** $tps(s)/tps(T[-1]) \leq 1 - \tau$ **then**
5:          $boundary \leftarrow (s_{\text{hi}}{=}T[-1],\ s_{\text{lo}}{=}s)$; **break**
6:      **end if**
7:      append $s$ to $T$; $cur \leftarrow cur/2$
8: **end while**
9: **if** $boundary \neq \bot$ **then**
10:      **if** $k$ is odd **then**
11:          select $\{s_{\text{hi}},\ \text{mid}(s_{\text{hi}}, s_{\text{lo}}),\ s_{\text{lo}}\} + (k-3)/2$ nearest on each side
12:      **else**
13:          select $\{s_{\text{hi}},\ s_{\text{lo}}\} + (k-2)/2$ nearest on each side
14:      **end if**
15: **else**
16:      $s^\star \leftarrow$ last element of $T$ (or $\max S$ if $T$ empty)
17:      **if** $k$ is odd **then** select $\{s^\star\} + (k-1)/2$ nearest on each side
18:      **else** select $(k/2)$ nearest on each side of $s^\star$
19:      **end if**
20: **end if**
21: **return** selected inputs (pair with fixed output anchors)

---

targets (e.g., effective memory bandwidth and peak arithmetic throughput of the SoC/GPU). Concretely, we parse device–backend–model metadata to obtain BW (bytes/s) and PK (FLOPs/s), and we parse model configuration to obtain hidden size $d$, number of layers $L$, attention heads $h$, KV heads $h_{kv}$, and feed-forward dimension $f$. Together with a parameter $\beta$ representing bytes per token, these quantities allow us to analytically compute the idealized runtime for any input/output length pair $(s, n)$:

$$T_{\text{roof}}(s,n) \;=\; \max\!\left(\tfrac{\text{FLOPs}(s,n)}{\text{PK}}, \tfrac{\text{Bytes}(s,n)}{\text{BW}}\right).$$

Here FLOPs$(s,n)$ and Bytes$(s,n)$ are closed-form functions derived from transformer structure and sequence lengths. This "analytical total ms" is then used either directly (linear regression on top of it) or in combination with model structure features to train predictors such as Roofline-LR and Roofline-RF. In this way, the predictor retains the physical interpretability of the roofline model while adapting it to heterogeneous mobile devices by injecting device-specific descriptors. appendx

**LR.** The core principle of SIMFLI is to decompose latency into two distinct phases: prefill and decode. Since these phases are affected differently by input and output token lengths, modeling them separately yields better predictive accuracy. As illustrated in Figure 3, prefill latency grows approximately linearly with the number of input tokens, while decode latency scales linearly with the number of output tokens. This separation has also been highlighted in prior work, motivating our design choice. Accordingly, we implement a simple yet effective baseline by selecting a small number of points from the token grid and applying linear regression (LR). This captures the intuitive linear relationship between input/output sizes and the corresponding phase-wise latency.

**Section-TPS.** For predicting the latency of LLMs on the server, Narayanan et al. (2023) propose a two–phase cost model that treats prefill and decode separately. They show that the end-to-end runtime can be written as the sum of a *piecewise-linear* function in prompt length (prefill) and a *linear* function in output tokens (decode). Concretely, they estimate section-wise throughput by first profiling runs with $o{=}1$ output token at multiple prompt sizes to obtain per-prompt coefficients $\{\alpha_j\}$ (the inverse of prefill TPS over each prompt range), then fitting a single slope $\beta$ from the linear relation of runtime w.r.t. $o{-}1$ to capture decode TPS. Combining these yields a per-(model, software, hardware) runtime predictor that generalizes across workloads and enables an "idealized runtime" metric for apples-to-apples comparison across stacks. :contentReference[oaicite:0]index=0

## 5.2 SETTINGS

As summarized in Table 3, we compare (i) model-LR, (ii) a roofline-based predictor (Imai et al., 2024), (iii) data-driven linear regression (LR), (iv) Section TPS (Narayanan et al., 2023), and (v) SimFLi.

For a fair comparison across methods, we uniformly restrict the measurement budget to five points per target. Input prompt lengths are chosen according to an exponential schedule, $\{8, 16, 32, 64, 120\}$, reflecting the practical need to span short- to long-context regimes with only a handful of samples. Output token lengths are selected randomly but held fixed across all methods by using a shared random seed, ensuring that each model is evaluated on identical $(n_{\text{in}}, n_{\text{out}})$ subsets. The only exception is SimFLi3, which incorporates its own Few-shot Point (FSP) selection algorithm. Rather than relying on the exponential schedule, SimFLi3 automatically identifies informative points from the token grid, thereby reducing sensitivity to arbitrary subset choices.

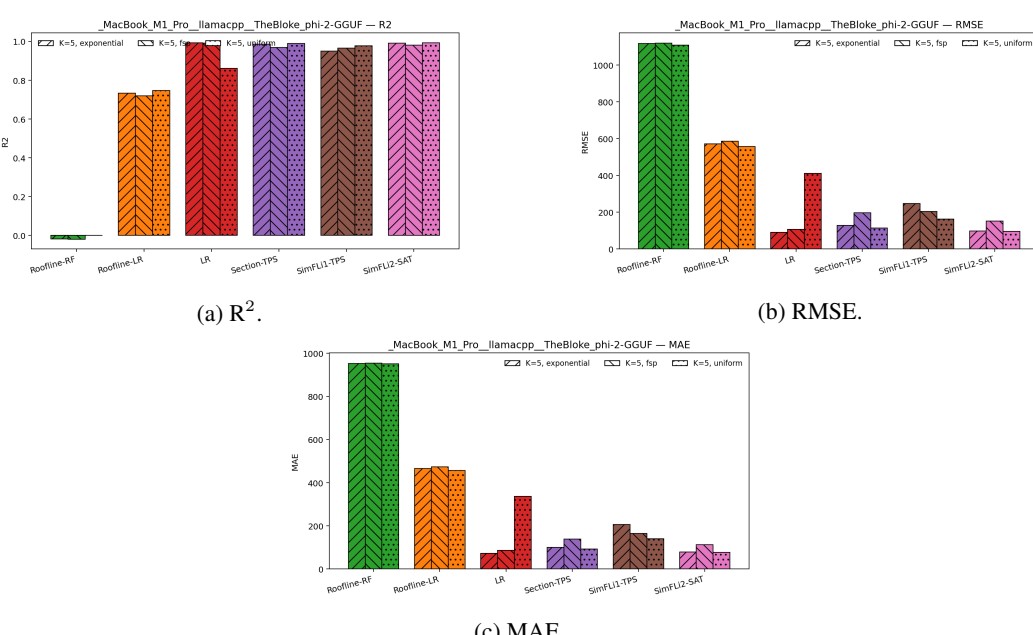

(a) $R^2$.  (b) RMSE.

(c) MAE.

Figure 4: Comparison of model performance metrics on
(M1 pro, llama.cpp, Phi-2)

## 5.3 RESULTS AND FINDINGS

We evaluate along two axes: (i) **in-distribution (ID) fit**—how well a method adapts to the given dataset for a fixed device–backend–model combination; and (ii) **within-combination cross-dataset generalization**—how well a model trained on one range of token grids transfers to a different range under the *same* combination (see Table 3 and Table 4).

**Main observations.** First, despite being trained with substantially more data, the roofline predictor exhibits large variance and underperforms simpler data-driven baselines at the same few-shot budget. In contrast, a heuristic-driven linear regression (LR) that reflects the empirical phase separation—prefill dominated by input length and decode dominated by output length—already outperforms the roofline model under a five-point budget.

Second, SECTION-TPS (Narayanan et al., 2023) surpasses SimFLi1 (TPS-only), highlighting a key limitation of SimFLi1: it does not explicitly model the *saturation* of prefill throughput observed in Figure 3. By incorporating a *saturation-aware* convergent fit for prefill TPS rather than relying on a few-point harmonic mean, SimFLi2 closes the gap to SECTION-TPS and achieves consistently high $R^2$ with competitive RMSE/MAE under on-device measurement constraints.

Third, SimFLi3 augments SimFLi2 with (a) an algorithm to approximate the saturation point and (b) a refined closed-form for prefill TPS conditioned on the estimated regime, while selecting measurement points via Few-shot Point selection (FSP). Despite using the same five-point budget, SimFLi3 matches the best methods (i.e., SimFLi2 and SECTION-TPS) on ID targets and maintains accuracy when evaluated on held-out token ranges within the same combination (Table 3 and Table 4), exhibiting low performance drop relative to ID.

**Takeaways.**  (i) Explicitly modeling *phase-wise* behavior with a *saturating* prefill component is essential for on-device profiling. (ii) Smooth "lining" in SimFLi2/SimFLi3 mitigates the brittleness of discrete sections in SECTION-TPS while retaining its strengths. (iii) With as few as five points, SimFLi achieves robust accuracy across ID and within-combination OOD settings, substantially outperforming traditional baselines under the same measurement budget.

Table 3: Baseline comparison (iPad10, llama.cpp, Qwen3-0.6B).
Exponential, 8_8_15_8_8_15 grid.

| Method | $R^2 \uparrow$ | RMSE $\downarrow$ | MAE $\downarrow$ |
|---|---|---|---|
| roofline-RF (Imai et al., 2024) | -0.0417 | 2297.4828 | 1967.4503 |
| roofline-LR (Imai et al., 2024) | 0.4473 | 1673.5035 | 1367.9353 |
| LR | 0.9554 | 58.8609 | 28.8811 |
| Section-TPS (Narayanan et al., 2023) | 0.9922 | 199.2662 | 117.6481 |
| SimFLi (LM) | 0.9821 | 301.0147 | 237.2067 |
| SimFLi (LM+SAM) | 0.9923 | 197.3643 | 99.1474 |
| SimFLi (LM+SAM+FSP) | 0.9886 | 240.6081 | 197.6322 |

Table 4: Baseline comparison (iPad10, llama.cpp, Qwen3-0.6b).
Test over 8_8_60_8_8_3 grid.

| METHOD | $R^2 \uparrow$ | RMSE $\downarrow$ | MAE $\downarrow$ |
|---|---|---|---|
| roofline-RF (Imai et al., 2024) | -29.4256 | 3149.8587 | 3097.6631 |
| roofline-LR (Imai et al., 2024) | -223.9937 | 11176.5956 | 7539.1492 |
| LR | -10.3778 | 8565.5679 | 1720.3516 |
| Section-TPS (Narayanan et al., 2023) | -6.8199 | 1926.1951 | 1313.6240 |
| SimFLi (LM) | -5.7777 | 1486.6701 | 1241.0823 |
| SimFLi (LM+SAM) | 0.9509 | 126.4209 | 109.5198 |
| SimFLi (LM+SAM+FSP) | 0.9730 | 93.8070 | 87.3347 |

# 6    CONCLUSION

In this work, we introduced **Simple Few-Shot Lining (SimFLi)**, a train-free and lightweight profiler for on-device LLM inference. SimFLi leverages the natural decomposition of latency into *prefill* and *decode* phases, and requires only a handful of token-length probes to estimate throughput and predict total latency. Unlike roofline predictors, which rely on server-centric assumptions, or token-grid regressions, which demand large numbers of target-specific measurements, SimFLi is directly aligned with prefill/decode characteristics and avoids the cost of training or device-specific calibration.

Our experiments across diverse devices, backends, and compact LLMs demonstrate that SimFLi achieves superior accuracy while requiring significantly fewer measurements. These results confirm that even with only a few carefully chosen points, SimFLi can reliably recover the full latency surface with high accuracy. This shows that accurate, device-tailored latency profiling does not need to be data-heavy or complex, but can be achieved with a simple few-shot approach. SimFLi thus provides a practical and deployable tool for mobile and edge environments.

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

# A  APPENDIX

## A.1  IMPLEMENTATION DETAILS

We found that increasing the context window beyond 512 or setting `prefill_chunk_size` greater than 128 in `MLC-LLM` frequently led to memory-related failures on our mobile devices (e.g., on iPad 10, whose RAM budget is tight enough that models near 2 GB become unstable). Consequently, we fixed the context length to 512 and set `prefill_chunk_size=128`. These choices align with commonly recommended defaults for mobile targets in the MLC-LLM stack and enabled stable runs across all devices and both backends considered in this study.

# B  LATENCY MEASUREMENT PROCEDURE

**Deterministic output length and sampling.**  To prevent early termination and ensure deterministic output lengths, we enabled `--ignore-eos` in `llama.cpp` and applied strong negative logit bias to EOS and stop tokens in `MLC-LLM`. Where supported, we also removed EOS/stop tokens from the tokenizer configuration to avoid accidental stops during decoding. All prompts were drawn from GSM8K (Cobbe et al., 2021). Tokenization used each backend's native tokenizer (MLC tokenizer for MLC-LLM; `llama.h`-based tokenizer for `llama.cpp`) to ensure consistency with runtime kernels.

**Repetition, cooldown, and device settings.**  Each latency configuration was repeated 5 times with a 5 s cooldown between runs to mitigate thermal throttling. During measurements, devices were set to maximum screen brightness, connected to Wi-Fi, and kept on charger power to minimize variation due to system-level power or brightness governors. We report per-point means over the repeated trials for all figures and metrics.

# C  MEMORY CONSTRAINTS

Several target devices (e.g., iPad 10 with 4 GB RAM) have modest memory headroom. A substantial portion is reserved by the OS and resident apps, leaving limited free memory for model weights, KV cache, and runtime buffers. In practice, models approaching or exceeding $\sim$2 GB (including runtime overheads) could trigger allocation failures or OS eviction under load. We mitigated this by using quantized variants (e.g., `q4` families in GGUF or `q4f16_1` in MLC), fixing the context window to 512, using `prefill_chunk_size=128` in MLC-LLM, and reusing model instances across runs whenever possible.

# D  ALTERNATIVE POINT SELECTION AND THRESHOLD

We adopt a few-shot point selection (FSP) strategy that exploits the saturation behavior of prefill throughput (tokens/s) with respect to input length $s$. Let $\mathrm{TPS}(s)$ denote the (mean) prefill tokens/s at input length $s$. Starting from $\min(\texttt{n\_ctx}, \max S)$, we repeatedly halve the target length and snap it to the nearest available $s$ (preferentially to powers of two). Between consecutive snapped points $s_{\mathrm{prev}}$ and $s_{\mathrm{cur}}$, we compute the ratio

$$\Delta = \frac{\mathrm{TPS}(s_{\mathrm{cur}})}{\mathrm{TPS}(s_{\mathrm{prev}})}.$$

When $\Delta \leq 1 - \tau$ (with threshold $\tau = \texttt{sat\_thresh}$, default 0.10), we mark $(s_{\mathrm{prev}}, s_{\mathrm{cur}})$ as the saturation boundary. For a budget of $K$ training pairs, we select $\{s\}$ by including the boundary endpoints and, if available, the midpoint in index space; the remaining slots are filled symmetrically around the boundary to cover both sub-saturated and saturated regimes. If no boundary is detected (e.g., the curve is nearly flat), we fall back to exponential spacing over the available $s$. Output lengths $\{n\}$ are sampled deterministically from the combo's available set with a fixed seed, and pairs $(s_i, n_i)$ are snapped to the nearest available grid point when necessary. This procedure matches the implementation in our code (`halving_trace_and_boundary`, `fill_sym`, and `k_exponential`) and is used consistently across experiments.

# E    LLM USAGE

We used an LLM to refine the sentences and ensure grammatical accuracy.

