# OpenReview forum: "SimFLi: Simple Few-Shot Linear Modeling for On-Device LLM Latency Profiling"
_ICLR.cc/2026/Conference — ICLR 2026 Conference Withdrawn Submission_

### Official Review · Reviewer_2knN · 2025-10-28

**Soundness:** 2
**Presentation:** 2
**Contribution:** 2
**Rating:** 2
**Confidence:** 3

**Summary:**

The paper proposes a simple few-shot linear modeling approach SimFLi for on-device LLM latency profiling. The method decomposes inference into prefill and decide phases, and estimates latency from only a few token-length probes. Finally, extensive experiments are conducted to show the estimation accuracy of the proposed method against existing competitors.

**Strengths:**

1. Estimating on-device latency is an important problem of practical use.
2. The proposed method is linear, which is simple. Also, it is training-free.
3. Experiments show the approximation accuracy of the proposed method.

**Weaknesses:**

1. The paper is not well written, which should be further polished.
2. The experiments are insufficient.
3. The contribution of the paper is incremental against existing works.

**Questions:**

1. I think the major problem of the paper lies in the limited contribution. For the preliminary observation that prefill and decode latency linearly scale with the input and output token lengths, it has already been characterized in the paper Narayanan et, al. (2023), as pointed out by the authors. Therefore, the real contribution of the paper is to propose linear formulations based on the observation. From the bar of ICLR, it is far beyond.
2. The authors claim that they have conducted wide range of experiments on heterogeneous hardware and backends. But in fact, only Galaxy represents Android which is not enough. Further, for Windows OS, there lack corresponding experiments. For LLMs, only Qwen and Phi are selected, which are also not enough.
3. The writing is another problem of the paper. For example, in Eq. 4, a, b and C appear without any explanation. In line 257, motivated by Figure 2d, but I think it is Figure 2. TPS appears in line 225 first, but there is not any explanation until line 259.
4. In Eq. 1, m is the number of shots. I am wondering how m affects the final results.

---

### Official Review · Reviewer_1RAc · 2025-10-30

**Soundness:** 2
**Presentation:** 3
**Contribution:** 2
**Rating:** 4
**Confidence:** 4

**Summary:**

This work presents SimFLi, a lightweight profiling method for predicting LLM inference latency on mobile and edge devices. The approach decomposes inference into prefill and decode phases and estimates latency using only a few token-length measurements, avoiding the need for extensive profiling or training that other related works do. SimFLi was evaluated against these existing approaches and consistently outperformed them in accuracy.

I find the topic of llm benchmarking very exciting and useful. While I enjoyed reading this work, I believe that the experimental methodology  has some issues (all mentioned below).

**Strengths:**

1. Interesting topic of LLM benchmarking. This area gets more and more important with the great and fast growth of LLMs, particularly on the mobile/edge side.
2. Easy to read paper, with useful background information summarized (e.g., prefill and decode phase, tps trajectories)
3. Great comparison with existing baselines
4. On-device experiments with both mobile platforms (doing in iOS is definitely a plus considering the existing automation limitations)
5. Overall the manuscript is very well written (with a few exception mentioned below)

**Weaknesses:**

1. The authors mentioned Edge devices, but this has only been tested in a MacBookPro, which is not a typical edge device, rather strong machine. I would have expected to see a RPI, or Jetson machine instead. This will also be compatible with other related works in the space.
2. As mentioned above, extending your evaluations to mobile devices is a big plus. However, a mobile device, unlike edge/desktops, is mainly used in mobile mode, and thus in discharging state. The experimented configuration is on charging mode that consequently has big performance benefits compared to the standard use of a phone.
3. Similar to the previous comment, a mobile phone under the pressure of llm inference is easily getting into a thermal throttling mode with significantly lower performance. 5sec pause in between runs is not enough to cool down and get out of that mode. These events can easily be captured in both iOS and Android.
4. While latency profiling is useful and interesting, it would have been better to also measure and report the implication on other metrics such as power, cpu and memory utilization.
5. Some related works that are not referenced:
  * MobileAIBench: Benchmarking LLMs and LMMs for On-Device Use Cases
  * MELTing point: Mobile Evaluation of Language Transformers
6. While llamacpp and MLC are quite popular, more frameworks are now available, and officially supported by big players like Google and Apple. For instance, MediaPipe by Google, use of MLX and run natively at OS level, PyTorch ExecuTorch etc. I would expect at least a reference and comment about why you haven't used these in your experiments.
7. Not all plots look good, especially on printed version. Font in the axis is too small, captions need to be fixed, some legends are not fully visible and too small (e.g., Figure 4a). Figure 1 is also too tiny (and could be moved at next page if possible).
8. Some (rare) grammatical issues here and there that should be fixed before acceptance (e.g., L079, sentence is not clear)

**Questions:**

1. Have you considered and applied SimFLi in edge devices like RPI or Jetson machines? If yes, were the results comparable with the ones reported in the manuscript?
2. What was the reason not configuring your devices into discharging mode? Also, would be good to add more details (possibly in the appendix) about the device configuration. For instance, was background data setting enabled? app updates? Adaptive Charging? Adaptive Battery? Adaptive brightness?
3. Have you measured and keep track of the thermal throttling events? I am quite certain that 5sec is not enough to restore the thermal state of the device.
4. Have you also tried to measure and report other metrics? To be clear, I understand that the focus is latency, and don't expect to consider other metrics into your formula. But I believe it would have been interesting and make the paper stronger if you could also report these and report the error.
5. Have you explored the other on device llm frameworks available, especially the ones from Apple and Google? If yes, what was the reason not including them into your experiments?

---

### Official Review · Reviewer_nbZQ · 2025-10-31

**Soundness:** 3
**Presentation:** 2
**Contribution:** 2
**Rating:** 2
**Confidence:** 4

**Summary:**

This paper introduces SimFLi, a training-free and lightweight method for on-device LLM latency profiling. It decomposes inference into prefill and decode phases and models latency as a simple linear or saturation-aware function of input/output token lengths. Unlike prior operator-level or NAS-based predictors, SimFLi estimates latency with only a few measurement points, avoiding data-heavy calibration or framework-specific hooks. Experiments across multiple mobile/edge devices and LLM backends (e.g., llama.cpp, MLC-LLM) demonstrate that SimFLi achieves high R² and low RMSE/MAE with as few as five measurements, outperforming roofline and regression baselines.

**Strengths:**

Simple yet effective methodology.
- The proposed decomposition into prefill and decode phases, together with few-shot linear modeling, is conceptually clear and computationally lightweight. Despite its simplicity, the method achieves high accuracy across different devices and frameworks.

Comprehensive empirical coverage.
- The experiments span multiple devices (smartphones, tablets, laptops) and two major backends (llama.cpp and MLC-LLM), demonstrating the method’s generality and low measurement overhead in realistic scenarios.

**Weaknesses:**

Insufficient motivation and unclear use case.
- The motivation for SimFLi is underdeveloped. The paper does not clearly explain why existing server-side latency predictors cannot be directly applied to on-device settings or what specific obstacles make them impractical. Moreover, it remains unclear why we need a fast few-shot profiling tool on mobile devices — the paper should explicitly describe realistic deployment or model-selection scenarios where SimFLi brings tangible benefits. Without such concrete context, the necessity of the proposed method feels underspecified.

Lack of consideration for GPU or NPU or heterogeneous deployment.
- Although the paper frequently refers to “heterogeneous devices,” all experiments are conducted on CPU-only platforms. No GPU or accelerator-based profiling is included, which limits the generality of the conclusions — particularly since many mobile frameworks (e.g., MLC-LLM, mllm) offload parts of computation to GPU or NPU.

Unclear writing and inconsistent notation.
- Several equations (e.g., Eq. (1)–(4)) introduce parameters without proper definition. The description of SimFLi1/2/3 variants is scattered, and figure captions lack sufficient detail. These issues make it difficult for readers to fully understand or reproduce the method. The paper would benefit from clearer mathematical notation and a more organized presentation.

**Questions:**

Same as weakness.

---

### Official Review · Reviewer_fRPB · 2025-10-31

**Soundness:** 3
**Presentation:** 4
**Contribution:** 3
**Rating:** 6
**Confidence:** 3

**Summary:**

The paper presents and evaluates a strategy for benchmarking performance of on-device models using device-tailored latency profiling in the proposed SimFli fewshot learning framework. The work has been evaluated on diverse set of hardware platforms and models against baseline profiling models.

**Strengths:**

- great evaluations on diverse devices and models
- simple setting and formulation for the estimator

**Weaknesses:**

- Model performance metrics could be affected by other factors when running on device (cacheing, paging, energy optimizations, etc)
- actual on-device models and novel models could have been evaluated (but I empathize this is a fast moving space)
- more focused hardware (with less external sensors and display etc such as jetsons or NPUs could be used (see papers in mobicom'24 and Mobicom'25 about this)

**Questions:**

I was wondering why the authors did not use more dedicated hardware for this assessment. Given the presence of dedicated on-device models and specialized hardware, this could have been useful.

What is the reason behind two decode phases in the method? Will there be much gains in performance?

Given the similarity of test profiles, would a teacher linear network not be beneficial in achieving the profiling once across different models?

nit: the legends and axes labels of figures such as fig 4 are too small to read.

---

### Note · Authors · 2025-11-13

I have read and agree with the venue's withdrawal policy on behalf of myself and my co-authors.